# The Consumption of a Synbiotic Does Not Affect the Immune, Inflammatory, and Sympathovagal Parameters in Athletes and Sedentary Individuals: A Triple-Blinded, Randomized, Place-bo-Controlled Pilot Study

**DOI:** 10.3390/ijerph19063421

**Published:** 2022-03-14

**Authors:** Carmen Daniela Quero-Calero, Oriol Abellán-Aynés, Pedro Manonelles, Eduardo Ortega

**Affiliations:** 1Faculty of Sport, Catholic University of Murcia, 30107 Murcia, Spain; cdquero@ucam.edu; 2International Chair of Sports Medicine, Catholic University of Murcia, 30107 Murcia, Spain; pmanonelles@ucam.edu; 3Grupo de Investigación en Inmunofisiología, Instituto Universitario de Investigación Biosanitaria de Extremadura (INUBE), University of Extremadura, 06006 Badajo, Spain; orincon@unex.es

**Keywords:** autonomic nervous system, immunity, inflammation, probiotics, sedentarism, soccer, synbiotics

## Abstract

This investigation aimed to identify the effect of a synbiotic in athletes and sedentary people, and their potential varying responses regarding the immune system, autonomic regulation and body composition. Twenty-seven participants were involved in the protocol: 14 sedentary and 13 semi-professional soccer players. Both groups were randomly divided into an experimental and control group. A synbiotic (Gasteel Plus^®^, Heel España S.A.U.) comprising a blend of probiotic strains, including *Bifidobacterium lactis* CBP-001010, *Lactobacillus rhamnosus* CNCM I-4036, and *Bifidobacterium longum* ES1, was administered to the experimental group, and a placebo was given to the control group for 30 days. Heart rate variability, body composition, and immune/inflammatory cytokines were determined. Statistically significant differences were observed between sedentary individuals and athletes in heart rate variability but not between the experimental and control groups. A difference between the athletic and sedentary group is observed with the influence of training on the effects of the synbiotic on the levels of fat mass and body-fold sum. No significant differences were shown in cytokines after the protocol study. No changes occur with the synbiotic treatment between the athlete and sedentary groups, while no negative effect was produced. Further research will be necessary to see chronic effects in the analyzed biomarkers.

## 1. Introduction

The heart rate variability (HRV) has traditionally been used to quantify cardiac autonomic activity [1] by applying it to different aspects such as health [2], psychological stress [3], or exercise-induced adaptations [4]. HRV is the study of the variability between consecutive R-waves of the electrocardiogram, and it has been firmly concluded that a greater variation between these consecutive waves is related to greater parasympathetic activity [5].

Since greater parasympathetic cardiac activity at rest is closely related to better health, it can be observed that HRV values are higher in healthy subjects than in subjects suffering from any disease [6]. Likewise, the effects of body composition can be highlighted as a factor that may intervene in such cardiac health as seen in obese individuals [7]. Physical training is also a determining factor in producing changes in HRV, since, in trained or athletic subjects, HRV is significantly higher than in less trained or sedentary subjects [8].

As mentioned in the previous paragraph, exercise can play a crucial role on HRV; other habits such as diet could be key aspects for the correct functioning of cardiac sympathovagal activity. Although there is not much literature on this subject, some studies have found results that provide information related to the fact that different dietary habits may be related to HRV values [9,10].

In addition, it is important to highlight the differences between amateur and professional athletes, since moderate and intense training predisposes more to suffer from cardiovascular problems, muscle damage, and infections [11,12,13], being nutrition a key factor for performance enhancement.

The influence of diet on health has been studied for decades, being necessary to determine some biomarkers that could identify those nutritional factors that could have a positive influence on long-term health [14]. Some examples include a Mediterranean diet, omega-3 fatty acids, polyphenols, and probiotics. Furthermore, some studies suggest that taking probiotics, prebiotics, and synbiotics can help athletes perform better by maintaining gastrointestinal and immunological function, lowering their susceptibility to disease [15,16,17], as well as improving some metabolic parameters due to changes in the microbiota composition and, consequently, in their body composition [18].

Inflammation is part of the immune response, and inflammatory responses could trigger a wide variety of diseases. Since the inflammatory response is related to the regulation of the autonomic nervous system (ANS), the biomarker HRV is given special attention in this study, as well as the importance of cytokines activation to prevent the release of inflammatory products into the bloodstream [19]. Therefore, synbiotics are proposed as a nutritional strategy for the improvement of the immune system and, as a consequence, possible improvements in the cardiovascular system.

Recent studies with the use of this synbiotic have concluded that it improves the quality of life associated with stress, anxiety, and sleep linked to improved immunoneuroendocrine interactions [20]. However, there are no specific studies concerning the effect of synbiotics on heart rate variability and its relation to the inflammatory/immune response, being this research pioneer in investigating whether there is a possible relationship between the uses of a synbiotic and possible improvements in the heart rate variability.

The main aim of the present investigation was to identify the effect of a nutritional supplement, a synbiotic, in athletes and sedentary people and their potential varying responses in relation to the immune/inflammatory system and body composition, as well as cardiovascular health.

## 2. Materials and Methods

### 2.1. Participants

A total of 27 participants, of which 14 sedentary students with low levels of physical activity (≤150 min/week) and 13 semi-professional soccer players of the Spanish football league, formed the final sample of the study. Two participants from the sedentary group were excluded before the start of the study, because they did not comply with the inclusion criteria, and “experimental death” occurred during the protocol due to an injury of the soccer players. Both groups were further divided randomly into two other groups: a group administered with the synbiotic and a control group that received a placebo. Random assignment through the use of coding provided by the laboratory allowed researchers and participants to be blinded to the treatment provided. The evaluation and analysis were also blinded, thus completing the triple-blind study.

### 2.2. Synbiotic

The nutritional supplement, the synbiotic Gasteel Plus^®^ from Heel España S.A.U. laboratories (Madrid, Spain), is made up of a mixture of probiotic strains and fructooligosaccharides (200 mg) as a prebiotic. Among the probiotic strains are: *Bifidobacterium lactis* CBP-001010, *Lactobacillus rhamnosus* CNCM I-4036, and *Bifidobacterium longum* ES. Each stick of said synbiotic contained lyophilized powdered bacteria, equivalent to ≥1 × 10^9^ colony-forming units (CFU), and also contained 1.5 mg of zinc, 8.25 µg of selenium, and 0.75 µg of vitamin D and maltodextrin as the excipient, the latter ingredient being the one used for the placebo sticks (300 mg). They were recommended to be taken in the morning, dissolved in water once a day, and stored at room temperature. The viability of the product was previously tested by Heel España S.A.U. laboratories.

### 2.3. Experimental Design

The main goal of this one-month, triple-blind, randomized, placebo-controlled pilot trial was to determine whether there were any differences in the effects of the synbiotic Gasteel Plus^®^ supplementation between sedentary people and athletes. Among the exclusion criteria were not suffering from any injury during the protocol and not consuming any type of probiotic, prebiotic, or fermented products (yogurt or other foods), as well as not taking any type of medication or supplement that could interfere with the results of the study. Participants were asked to follow their regular diet two weeks prior to the investigation and during the protocol. The soccer players followed the diet prepared by their nutritionist.

All participants were informed about the study two weeks before the intervention and asked to provide written informed consent before participating in the study, which had previously been approved by the ethics committee of the Catholic University of Murcia (Spain) in accordance with the current legislation (CE031810). ClinicalTrials.gov was used to register this trial (identifier: NCT04776772: available from website). This research is part of a larger study, and there is a previously published article with the same sample [20].

The participants had to come to the laboratory twice, once at the baseline, prior to ingestion of the synbiotic or placebo, and again 30 days later. The procedures and materials of the tests were performed in the same way for the “baseline tests” and “final tests” to alter each of the measurements as little as possible, and participants had fast for at least 12 h prior to sampling.

### 2.4. Anthropometry

All anthropometric variables were performed by a level 1 or level 2 anthropometrist certified by the International Society for the Advancement of Kinanthropometry (ISAK). Likewise, all measurements were performed following the ISAK protocols on the right side of the body.

Body mass was determined using a digital scale SECA 862 (SECA, Hamburg, Germany); height was assessed with a GPM anthropometer (Siber-Hegner, Zurich, Switzerland); circumferences with a Lufkin non-stretchable metal tape Lukfin W606PM (Lufkin, MO, USA); and the skinfold thickness measured at 8 sites: triceps, subscapular, biceps, iliac crest, supraspinal, abdomen, thigh, and mid leg with a Harpenden skinfold caliper (British indicators Ltd., Burgees Hill, UK). After retrieving the data of anthropometric measurements, the body composition was estimated.

The percentage of muscle mass was analyzed by the equation proposed by Lee et al. [21], and the percentage of fat mass was estimated according to the equation of Withers et al. [22]. To avoid errors, all instruments were calibrated prior to taking the measurements. Each variable was taken twice or three times, if the difference between the first two measurements was greater than 5% for the folds and greater than 1% for the rest of the variables recorded, with the use of mean or median values, respectively, for the subsequent statistical analysis. Furthermore, the sum of the 6 upper-body skinfolds (6Sk) was calculated, as well as the sum of the 8 skinfolds (8Sk).

### 2.5. Heart Rate Variability

The HRV measurements were carried out at resting in a supine position for 10 min, being analyzed only the last 5 min to ensure that a resting heart rate was achieved. The analysis of only 5 min were done in accordance with Camm et al. [23].

HRV estimations were performed with a pulse sensor Polar H7 (Polar Electro Ltd., Kempele, Finland) to assess beat-to-beat data during the evaluations. The examination of HRV variables was done through the product Kubios HRV 3.0. This product was additionally used to apply filters for artifact expulsion if necessary. The time and frequency domains, as well as Poincare plot variables, were recovered. The time domain factors investigated were the mean heart rate (HR), mean R–R interval time in ms (RR), the standard deviation of consecutive R–R intervals (SDNN), the root mean square of successive differences of consecutive R–R intervals in ms (RMSSD), and the relative value of consecutive intervals that differ by more than 50 ms (pNN50). Additionally, the Fast Fourier transformation was utilized to analyze the spectral components of the frequency domain. The high recurrence power (HF) (0.15–1.0 Hz) and low recurrence power (LF) (0.04–0.15 Hz) segments were determined as integrals of the particular force otherworldly thickness bend. These factors were communicated in normal logarithm changed qualities (HFln and LFln separately). Thus, the proportion LF/HF was evaluated as well. At last, the Poincare plot factors, for example, the standard deviation of the prompt beat-to-beat RR stretch inconstancy (SD1) and the standard deviation of persistent long-haul R–R-span changeability (SD2), were determined. The stress score (SS) was broken down by the condition 1000 × 1/SD2, and the thoughtful/parasympathetic proportion (S/PS) was determined through SS/SD1.

### 2.6. Blood Samples

Blood samples were taken from the individuals at 8 a.m. and deposited into collection tubes containing the anticoagulant EDTA and coagulating agents, respectively, to isolate the plasma and serum. The plasma and serum were centrifuged for 10 min at 1600 and 1800× *g*, respectively. As the serum and plasma samples were collected, they were tagged and gradually refrigerated at −20 °C. Finally, samples were kept at −80 °C until they could be analyzed.

The LuminexTM 200 System instrument (Luminex Corporation, Austin, TX, USA) was utilized to determine the examined cytokines, Interleukin 6 (IL-6), Interleukin 8 (IL-8), and Tumor necrosis factor alpha (TNF-α) using the ProcartaPlex TM Multiplex Immunoassay. The procedures were carried out according to the manufacturers’ instructions, and the results were quantified using an ELISA auto analyzer (Sunrise, Tecan, Männendorf, Switzerland).

### 2.7. Statistical Analysis

The SPSS for Windows statistical tool was used for data collection, treatment, and analysis (version 20.0; SPSS, Inc., Chicago, IL, USA). The mean and SD of the descriptive statistics were calculated. The Shapiro–Wilk test was used to check the assumption of normality before performing parametric testing. It was used in a two-way analysis of variance to investigate the main effects and the interactions between the group factor (athlete vs. sedentary), intake factor (synbiotic vs. placebo), and time factor (pre-intervention vs. post-intervention), as well as the presence of differences between groups in the outcomes. The eta squared partial (η2p) was also calculated to assess the effect size of the comparisons. A level of *p* ≤ 0.05 was set to indicate statistical significance.

## 3. Results

Table 1 shows the anthropometric characteristics of the sample. No significant differences (*p* > 0.05) were observed between groups in any of the descriptive variables.

### 3.1. Heart Rate Variability

Table 2 shows the HRV data obtained in the sample. The main effects after the intervention are very low, so that no significance is observed in any of the HRV variables in the effect of time. Thus, no significant changes were found in any of the HRV variables in any of the groups. Similarly, the effects of intake showed similar results with very low interactions and no significance. In the intergroup effect, differences were observed when comparing the athlete group with the sedentary group, with some high interaction values in variables such as HR or pNN50.

### 3.2. Anthropometry

Table 3 shows the data of the anthropometric variables where the interaction values of the three main effects are observed. Regarding the interaction of time, weight and variables related to subcutaneous fat show high interactions and significance, observing that the most outstanding changes occur in the synbiotic sedentary group. The interaction of group presents high values and significance in the variables related to subcutaneous fat and muscle mass, where it is seen that the difference between athletes and sedentary individuals is maintained from the baseline to after the intervention. However, these types of results are different in the interaction with the intake, since the F values are very low, and no significance is shown between groups.

Figure 1 shows the results of the inflammatory regulator IL-6 (both pro- and anti-inflammatory, depending on the context of the immune response). The time effect presented values of F = 1.049; *p* = 0.316 and η^2^p = 0.044, group effect showed F = 16.979; *p* < 0.001 and η^2^p = 0.425, and an intake interaction of F = 0.015; *p* = 0.904 and η^2^p = 0.001.

### 3.3. Blood Samples

Figure 2 and Figure 3 show the results of the circulating concentration of proinflammatory cytokines (IL-8, TNF-α) showing that the group of athletes presented, like the sedentary individuals, basal levels in normal ranges of inflammatory response. IL-8 effects showed values of F = 2.112; *p* = 0.16 and η^2^p = 0.084 for the time effect, F = 1.12; *p* = 0.301 and η^2^p = 0.046 for group effect, and F = 0.01; *p* = 0.921 and η^2^p < 0.001 for the intake effect. It was observed a statistically significant change (*p* < 0.05) in the time effect in the athlete-placebo group. Regarding TNF-α it was observed an interaction effect of F = 1.587; *p* = 0.22 and η2p = 0.065 for the time effect, F = 0.332; *p* = 0.57 and η^2^p = 0.014 for group effect, and F = 0.792; *p* = 0.383 and η^2^p < 0.33 for the intake effect.

## 4. Discussion

### 4.1. Synbiotic Effects on Heart Rate Variability

In the Results section, it was shown that the resting heart rate was significantly lower in the athletic population than in sedentary subjects. This fact was previously known, as it is one of the main adaptations to sports training thanks to a better efficiency of the heart beat [24] due to the hypertrophy of the left ventricular wall and other factors [25]. Regarding HRV, it is also observed that the inter-beat variability is higher in the athlete sample when compared to sedentary population; this cardiac variability is the result of the interactions between the autonomic nervous system and the cardiovascular system. This fact can be observed in the SDNN, as this variable gives us a direct value of the overall variability throughout the resting measurement [26]. In addition, pNN50 is also significantly higher in soccer players, indicating that, in the short term, the heartbeat is less predictable, which is also related to adaptation to physical exercise [27]. RMSSD is directly related to vagal tone [2]; therefore, it can be observed that there is less parasympathetic activity at rest in the sedentary sample. It has been previously observed that sedentary subjects show lower RMSSD values when compared to trained athletes [28]; this fact is also explained as an adaptation to high-intensity training [29], as it seems to help in the regulation of the sympathovagal balance. However, we did not observe changes in resting sympathetic activity between athletic and sedentary subjects measured by SS.

The sedentary sample did not undergo HRV-related adaptations, which has been observed previously in other studies [30], as no adaptations to physical exercise occur. The use of a synbiotic did not prove to be effective in inducing adaptations in cardiac sympathovagal activity; however, it should be noted that the intervention period was too short to observe this type of adaptations, as they need longer periods to occur. In the sports population, we observed the same lack of adaptations in the synbiotic and placebo groups. Regarding the contribution that some dietary supplements could have in improving cardiovascular function, it has been shown that the gut microbiota is involved in blood pressure control through various mechanisms, such as exerting control at the autonomic and central nervous system levels as endothelial protection function [31], as well as in the improvement of triglyceride and cholesterol levels after the use of probiotics [32]. However, observing the results obtained in this study, it cannot be concluded that significant changes in heart rate variability in athletic and sedentary subjects are produced after ingestion of the synbiotic over a 30-day period. It would therefore be interesting for future research to analyze longer periods in order to observe the desired effects.

### 4.2. Synbiotic Effects on Body Composition

According to Clemente et al. [33], the regular participation of young people in a sporting activity, thus increasing the levels of physical activity and decreasing the levels of sedentary lifestyle, would be a key factor in the decrease of body weight. However, in this study, no significant differences were observed in the baseline situation between sedentary and athletic subjects in reference to weight and body mass index. However, the results of this study show some significant differences in body weight (Table 2), as well as in body mass index (BMI), in the athletic participants who consumed placebo after the treatment period, with a significance level of *p* < 0.05.

There are some studies showing changes in reference to weight and BMI in subjects who consumed a synbiotic (*Lactobacillus salivarius* LS33) for a period of 30 days [34]. The impact of probiotic consumption on different parameters related to obesity and overweight has also been studied, showing a reduction in body weight in mice fed a high-fat diet together with several probiotic strains [35]. Another study with healthy adults and the use of a probiotic (*L. plantarum* TENSIA) also showed healthier BMI values after the intervention [36]. To date, there have been few studies conducted on humans to examine the possible effect that probiotics, prebiotics, and/or synbiotics might have on body weight and BMI. A recent study [18] indicated that there were no statistically significant differences in the body composition parameters and obesity-related biomarkers between the placebo and synbiotic groups at the end of a clinical trial using different probiotic strains of *Lactobacillus acidophilus* DDS-1, *Bifidobacterium lactis* UABla-12, *Bifidobacterium longum* UABl-14, and *Bifidobacterium bifidum* UABb-10 and a trans-galactooligosaccharide (GOS) as the prebiotic component.

In reference to the notable differences in anthropometric measurements, there is strong evidence that the percentage of muscle mass is generally higher in athletic subjects than in sedentary individuals, while the percentage of fat mass and, therefore, the sum of six and eight folds is lower in athletic subjects [37]. This is shown in our study (Table 2), in which significant differences (*p* < 0.05) are observed with respect to baseline measurements between sedentary individuals and soccer players.

A change in the “statistical interaction” (*p* < 0.05) between the athletic and sedentary group is shown with the influence of training on the effects of the synbiotic on the levels of fat mass and body-fold sum, where the sedentary group increases significantly after treatment, while, in the athletic group, differences are only observed in the placebo group. This may be due to the effect of the sport itself [38,39], together with the possible effect of the synbiotic.

Some probiotic strains have been shown to be influential in reducing body weight, as well as fat reserves, in humans [40]. In addition, some genera of bacteria, such as *Bifidobacterium*, have acted in other areas of the human body, such as the liver, where they have influenced the decrease in lipid reserve, with a consequent reduction in body weight and body fat [41]. It has been shown that certain amounts of conjugated linoleic acid in adipose tissue, contained in certain probiotic strains, can reduce body mass in mice and humans [42]. Some authors [43] demonstrated that the consumption of a strain of *Lactobacillus paracasei* in mice was associated with an increase in angiopoietin type 4, an inhibitor of lipoprotein lipase that controls the deposition of triglycerides in adipocytes. Again, these results depend on the type of strain used in each investigation and are not general to all studies.

No major changes can be attributed to the consumption of the synbiotic, but the intake of the synbiotic had no negative effect on some of the parameters studied, such as the muscle mass of the athletes. In addition, according to the levels of fat mass, this synbiotic treatment in athletes was able to prevent the increase in fat mass during the protocol, both in sedentary subjects and in athletes. In any case, it would be plausible to think that these effects, if any, would occur in overweight rather than normal weight individuals, in whom variations in this sense could even be considered “unhealthy”.

### 4.3. Synbiotic Effects on Immune/Inflammatory System

The main objective of an athlete’s training program is to improve the performance. In this attempt to improve, athletes increase their exercise levels and intensity, which can generate physical and mental stress that could create poor physiological adaptations, as well as possible affections in the immune system [15,44]. High musculoskeletal stress results in muscle tissue damage and systemic inflammatory response, which is transmitted through inflammatory chemical signals from the immune system, such as cytokines [45]. In line with Smith [46], inflammatory tissue trauma results in the production of a number of proinflammatory cytokines (IL-8, IL-6, and TNF-α), which leads to the development of illness or fatigue in the athlete’s behavior.

Probiotics, prebiotics, and synbiotics could be a good strategy to improve human health through the immunomodulation of local immunity either by maintaining the integrity of the intestinal wall or by acting on systemic immunity [44]. There are a multitude of studies, mostly with the use of probiotics, that have studied the possible interactions of such supplements on the inflammatory/immune system [47,48,49,50,51,52,53] and others, although to a lesser extent, with synbiotics [54,55,56].

According to our previous results [20], the synbiotic intervention clearly improved sleep quality, as well as perceived general health, stress, and anxiety levels, in the athlete group, inducing an immunophysiological bioregulatory effect, depending on the basal situation of each experimental group.

The increase of IL-6 in the circulation during exercise appears to be the cause of the subsequent increase in concentrations of the anti-inflammatory cytokine IL-10 and the antagonistic IL-1 receptor, which stimulate the release of cortisol (a hormone that suppresses the immune system) by the adrenal glands [57]. 

No significant differences were found in the levels of the cytokine IL-6 in our study (Figure 1), coinciding with a recent investigation [58] that completed a probiotic treatment conducted on healthy sedentary subjects. This, in turn, agrees with the results in another investigation [53], where 23 male athletes were involved in which IL-6 had no significance according to treatment. In addition, in harmony with one recent review and meta-analysis study [59], there were no statistically significant changes in reference to IL-6 and IL-8 cytokines in healthy subjects, with the possible variations being more accentuated in the use of synbiotics versus probiotics. An increase in the levels of the cytokine IL-8 is shown in our study in the placebo group of athletes. These results could be due to training, and given the very proinflammatory nature of this cytokine, the synbiotic could attenuate the inflammatory response to training during the stressful period of competition.

As the results of this study show in Figure 3, no significant differences were observed in the levels of TNF-α after the study intervention, coinciding with the results of one study [50,51] in which supplementation with *L. casei* for a period of 7 days did not induce significant changes in some cytokines that were analyzed (TNF-α, IL-8, IL-1β, IL-6, and IL-10). These studies are contradictory to others in which significant changes in some of the aforementioned cytokines do occur. Some authors [52] stated that changes in plasma circulating values of the cytokines TNF-α, IL-1β, and IL2 occurred after consumption of a probiotic *Bacillus coagulans* combined with calcium, and as defined in other study [53], the cytokine TNF-α showed similar basal levels, while, after 14 weeks of probiotic treatment, it significantly decreased in the experimental group.

There are controversies among the results of the various studies due to the different uses of bacterial strains, as well as the type of population under study. It is necessary, in order to obtain future and more concise results, for a greater methodological unification, as well as an optimal selection of the functional markers to be determined.

One of the most important limitations of the study is the low sample size in each group. Since convenience sampling was used in order to use a sample of high-level athletes, the sample of sedentary subjects was obtained from the sample of athletes in order to have the same number of participants. Therefore, an important limitation of this study was not to use a calculation to obtain a representative sample based on the total population, which would have resulted in a higher statistical power. Another important limitation was that it was not possible to control the diet of the sedentary sample, who followed their normal daily diet.

## 5. Conclusions

In summary, synbiotics appear to have no effect on heart rate variability, as well as on some inflammatory markers after use for a short period of 30 days in healthy sedentary individuals and athletes. The effects observed on various parameters of body composition could be due both to the effect of the exercise itself and to the possible influences of the synbiotic. In either case, the consumption of the synbiotic had no negative or detrimental effects, so more research is needed on this topic, with improved methodological unification and for a longer study time in order to observe not only acute but also chronic changes.

## Figures and Tables

**Figure 1 ijerph-19-03421-f001:**
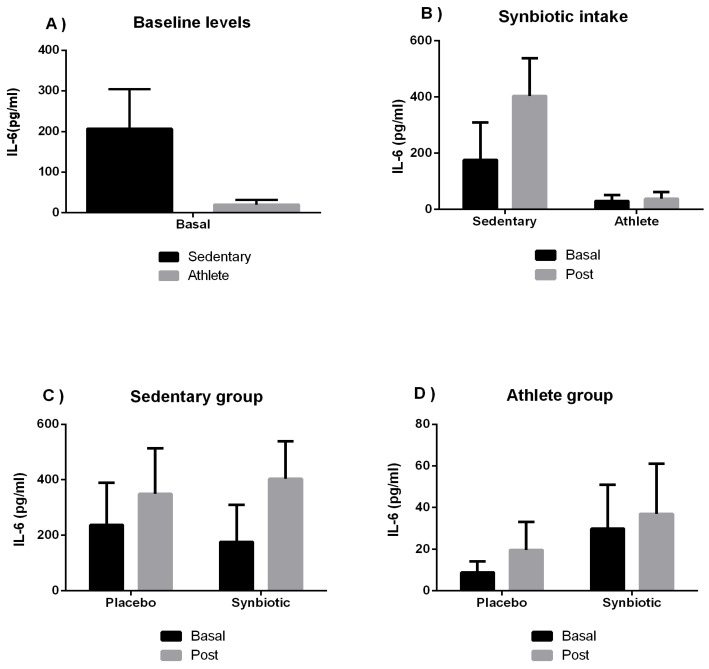
The effect of training and a synbiotic on the IL-6 cytokine. (**A**) Baseline serum IL-6 concentrations in sedentary men (n = 14) and athletes (n = 13). (**B**) Training effects on synbiotic effects on serum IL-6 concentration (n = 7 and n = 6 in sedentary and athlete groups, respectively). (**C**) Effect of synbiotic consumption on IL-6 in sedentary individuals with placebo (n = 7) or synbiotic (n = 7). (**D**) Effect of synbiotic consumption on IL-6 in athlete individuals with placebo (n = 6). The determinations are expressed by the mean ± SD of each of the samples. No significant differences were seen between groups (*p* > 0.05).

**Figure 2 ijerph-19-03421-f002:**
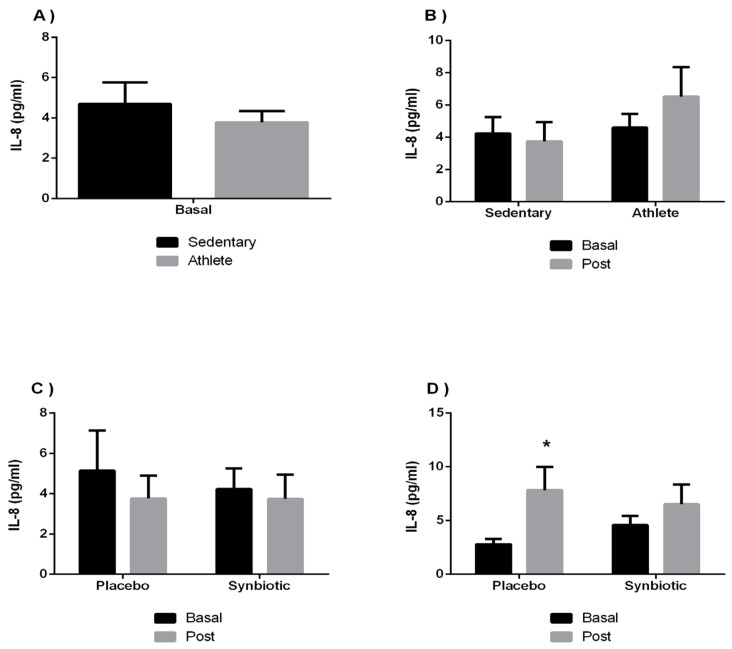
The effect of training and a synbiotic on the IL-8 cytokine. (**A**) Baseline serum IL-8 concentrations in sedentary men (n = 14) and athletes (n = 13). (**B**) Training effects on synbiotic effects on serum IL-8 concentration (n = 7 and n = 6 in sedentary and athlete groups, respectively). (**C**) Effect of synbiotic consumption on IL-8 in sedentary individuals with placebo (n = 7) or synbiotic (n = 7). (**D**) Effect of synbiotic consumption on IL-8 in athlete individuals with placebo (n = 6). The determinations are expressed by the mean ± SD of each of the samples. * *p* < 0.05 compared to basal.

**Figure 3 ijerph-19-03421-f003:**
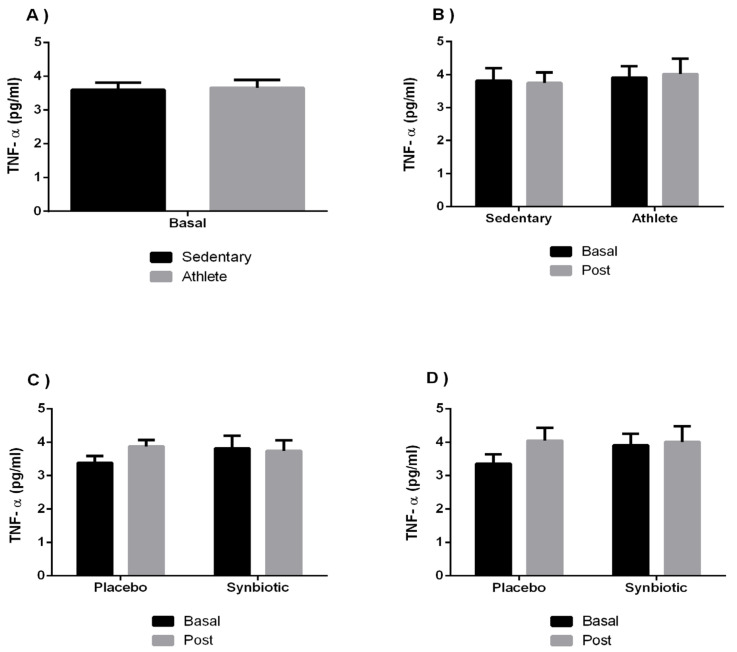
The effect of training and a synbiotic on the TNF-α cytokine. (**A**) Baseline serum TNF-α concentrations in sedentary men (n = 14) and athletes (n = 13). (**B**) Training effects on synbiotic effects on serum TNF-α concentration (n = 7 and n = 6 in sedentary and athlete groups, respectively). (**C**) Effect of synbiotic consumption on TNF-α in sedentary individuals with placebo (n = 7) or synbiotic (n = 7). (**D**) Effect of synbiotic consumption on TNF-α in athlete individuals with placebo (n = 6). The determinations are expressed by the mean ± SD of each of the samples. No significant differences were seen between groups (*p* > 0.05).

**Table 1 ijerph-19-03421-t001:** Descriptive data of the sample.

	Athlete		Sedentary	
Variable	Synbiotic (*n* = 7)	Placebo (*n* = 6)	Synbiotic (*n* = 7)	Placebo (*n* = 7)
Age (years)	20.6± 1.39	21.9 ± 2.77	23.04 ± 2.09	24.31 ± 3.94
Body mass (Kg)	70.57 ± 6.75	73.95 ± 6.42	77.47 ± 13.47	79.81 ± 8.05
Height (cm)	178.23 ± 4.78	180.6 ± 8.57	176.23 ± 4.49	183.97 ± 7.3

Results are shown as the mean ± standard deviation.

**Table 2 ijerph-19-03421-t002:** Heart rate variability before and after the consumption of a synbiotic or placebo in athletes and sedentary individuals.

								ANOVA (F, *p*, η^2^_p_)								
			Pre		Post			Time Effect			Group Effect			Intake Effect		
Outcome	Group	Intake	M	SD	M	SD	*p*	F	*p*	η^2^_p_	F	*p*	η^2^_p_	F	*p*	η^2^_p_
HR (bpm)	Sedentary	Placebo	70.75	11.83	59.51	13.84	0.461	0.263	0.613	0.011	18.02	<0.001	0.439	0.286	0.598	0.012
		Synbiotic	77.46	5.78	80.30	15.03	0.532									
	Athlete	Placebo	59.64	11.55	61.06	10.90	0.772									
		Synbiotic	59.51 *	13.84	56.58 *	9.38	0.518									
SDNN (ms)	Sedentary	Placebo	88.10	39.53	97.84	18.95	0.354	<0.001	0.995	<0.001	4.910	0.037	0.176	0.818	0.375	0.034
		Synbiotic	60.46	25.65	57.79	23.97	0.832									
	Athlete	Placebo	81.35	17.07	100.48	49.64	0.168									
		Synbiotic	97.84 *	18.95	93.31 *	24.14	0.720									
pNN50 (%)	Sedentary	Placebo	22.19	19.48	45.46	21.87	0.811	0.025	0.877	0.001	16.173	0.001	0.413	0.014	0.907	0.001
		Synbiotic	9.87	8.73	10.89	11.11	0.874									
	Athlete	Placebo	35.68	19.68	37.17	18.70	0.830									
		Synbiotic	45.46 *	21.87	46.51 *	19.13	0.869									
RMSSD (ms)	Sedentary	Placebo	52.41	36.43	75.41	33.74	0.633	0.244	0.626	0.01	9.585	0.005	0.294	0.049	0.826	0.002
		Synbiotic	32.00	13.18	31.73	14.47	0.982									
	Athlete	Placebo	62.35	24.76	73.30	42.24	0.401									
		Synbiotic	75.41 *	33.74	75.83 *	29.86	0.972									
HF_ln_ (ms^2^)	Sedentary	Placebo	6.25	1.50	6.86	1.35	0.502	0.218	0.645	0.009	5.324	0.03	0.188	0.057	0.814	0.002
		Synbiotic	6.01	0.68	5.89	0.75	0.715									
	Athlete	Placebo	6.77	0.85	7.22	1.01	0.241									
		Synbiotic	6.86	1.35	7.10	0.92	0.489									
LF/HF	Sedentary	Placebo	4.71	3.15	3.45	4.28	0.61	0.136	0.715	0.006	0.641	0.431	0.027	0.107	0.746	0.005
		Synbiotic	2.90	1.98	3.16	2.09	0.835									
	Athlete	Placebo	2.84	2.54	2.39	1.91	0.741									
		Synbiotic	3.45	4.28	3.34	3.00	0.931									
SS (Hz)	Sedentary	Placebo	10.08	4.73	8.18	1.83	0.164	1.111	0.303	0.046	4.894	0.037	0.175	0.408	0.53	0.017
		Synbiotic	14.03	5.63	15.83	10.09	0.425									
	Athlete	Placebo	9.78	1.78	8.81	3.24	0.689									
		Synbiotic	8.18 *	1.83	8.94	2.75	0.736									
S/PS	Sedentary	Placebo	0.55	0.71	0.20	0.14	0.249	1.354	0.257	0.056	5.531	0.028	0.194	0.108	0.745	0.005
		Synbiotic	0.76	0.48	1.10	1.43	0.225									
	Athlete	Placebo	0.27	0.14	0.25	0.21	0.959									
		Synbiotic	0.20 *	0.14	0.20	0.11	1									

HR: Heart rate; SDNN: Standard deviation of consecutive R-R intervals; pNN50: relative value of consecutive intervals that differ by more than 50 ms; RMSSD: root mean square of successive differences of consecutive R-R intervals; HF_ln_: High-frequency power based on its natural logarithm; LF/HF: Ratio between low- and high-frequency power; SS: Stress Score; S/PS: sympathetic–parasympathetic ratio. * *p* < 0.05 compared to Sedentary.

**Table 3 ijerph-19-03421-t003:** Anthropometric values before and after the consumption of a synbiotic or placebo in athletes and sedentary individuals.

								ANOVA (F, *p*, η^2^_p_)								
			Pre		Post			Time Effect			Group Effect			Intake Effect		
Outcome	Group	Intake	M	SD	M	SD	*p*	F	*p*	η^2^_p_	F	*p*	η^2^_p_	F	*p*	η^2^_p_
Body Mass (Kg)	Sedentary	Placebo	79.81	8.05	79.97	8.32	0.698	8.529	0.008	0.271	2.981	0.098	0.115	0.571	0.458	0.024
		Synbiotic	77.47	13.47	78.24	14.05	0.066									
	Athlete	Placebo	73.95	6.42	74.73	5.95	0.083									
		Synbiotic	70.57	6.75	71.24	7.42	0.107									
6_Sk_ (mm)	Sedentary	Placebo	86.11	29.38	91.39	29.97	0.27	11.451	0.003	0.332	20.46	<0.001	0.471	0.932	0.344	0.039
		Synbiotic	107.41	58.99	127.43	65.38	<0.001									
	Athlete	Placebo	40.25 *	6.49	45.73 *	8.90	0.289									
		Synbiotic	39.74 *	4.70	41.27 *	11.27	0.747									
8_Sk_ (mm)	Sedentary	Placebo	109.94	37.55	118.68	41.29	0.107	18.778	<0.001	0.449	19.772	<0.001	0.462	0.935	0.344	0.039
		Synbiotic	138.50	74.07	166.26	86.27	<0.001									
	Athlete	Placebo	52.88 *	9.33	60.50 *	11.81	0.189									
		Synbiotic	51.87 *	5.35	53.80 *	15.25	0.714									
Fat (%)	Sedentary	Placebo	16.05	5.44	16.93	5.55	0.311	13.962	0.001	0.378	18.506	<0.001	0.446	1.019	0.323	0.042
		Synbiotic	20.15	11.26	24.47	13.44	<0.001									
	Athlete	Placebo	7.70 *	1.08	8.62 *	1.55	0.321									
		Synbiotic	7.56 *	0.82	7.87 *	1.97	0.721									
Muscle (%)	Sedentary	Placebo	40.44	4.61	40.55	4.63	0.917	0.362	0.553	0.016	8.662	0.007	0.274	0.002	0.966	<0.001
		Synbiotic	40.91	6.23	38.49	7.14	0.043									
	Athlete	Placebo	44.20	1.97	45.60	5.72	0.264									
		Synbiotic	45.79 *	2.45	45.30 *	2.08	0.669									

6_Sk_: Sum of 6 upper-body skinfolds. 8_Sk_: Sum of 8 skinfolds. * *p* < 0.05 compared to Sedentary.

## Data Availability

Not applicable.

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
