# Peer review of "The Consumption of a Synbiotic Does Not Affect the Immune, Inflammatory, and Sympathovagal Parameters in Athletes and Sedentary Individuals: A Triple-Blinded, Randomized, Place-bo-Controlled Pilot Study"

_ijerph, 2022, doi:10.3390/ijerph19063421_

Round 1
Reviewer 1 Report
Does the consumption of a synbiotic affect to immune, inflammatory and sympathovagal parameters in athletes and sedentary individuals? A Triple-Blinded, Randomized, Placebo-Controlled Pilot Study.
Quero-Calero et al aimed to identify the effect of a nutritional 65, a synbiotic, in athletes and sedentary people, and their potential varying responses in relation to the immune/inflammatory system, body composition as well as cardiovascular health.
The paper is interesting but there are two weakness that threaten the validity of the results.
- The protocol was registered on March 2, 2021 but the study started on May 15, 2019. Then, this is a retrospective Randomized Clinical Trial. In the registration there is no mention about a pilot study.
- In the paper is not stated what parameters have been set to calculate the sample size. In this way it Is not possible to interpret any result above all, considering that the protocol has been registered with 42 primary outcomes measures.
I think authors would have to make major revisions to make the paper suitable for publication.
Author Response
Quero-Calero et al aimed to identify the effect of a nutritional 65, a synbiotic, in athletes and sedentary people, and their potential varying responses in relation to the immune/inflammatory system, body composition as well as cardiovascular health.
Firstly, we would like to thank you for your time and effort in evaluating the manuscript. Thank for your positive considerations, and we sincerely appreciate all of your helpful, detailed comments and suggestions.
The paper is interesting but there are two weakness that threaten the validity of the results.
- The protocol was registered on March 2, 2021 but the study started on May 15, 2019. Then, this is a retrospective Randomized Clinical Trial. In the registration there is no mention about a pilot study.
Thanks for the appreciation, we didn't realize it during registration. We have considered calling the pilot study because it is part of a larger study with more participants that has not yet been carried out but will be conducted to increase the sample and obtain more accurate results.
- In the paper is not stated what parameters have been set to calculate the sample size. In this way it is not possible to interpret any result above all, considering that the protocol has been registered with 42 primary outcomes measures
We did not calculate de sample size. We are aware of this limitation, therefore we have included a limitations paragraph in the discussion section explaining why we did not calculate the representative sample size since we used a convenience sampling. The information described is:
“Since convenience sampling was used in order to use a sample of high-level athletes, the sample of sedentary subjects was obtained from the sample of athletes in order to have the same number of participants. Therefore, an important limitation of this study was not to use a calculation to obtain a representative sample based on the total population, which would have resulted in a higher statistical power.”
Reviewer 2 Report
Aim of this RCT is to identify the effect of a synbiotic in athletes and sedentary people, and their potential varying responses regarding the immune system, autonomic regulation and body composition.
Many studies have been published on the subject with poor results. The same authors have also recently published a paper very similar to this one. Many parts turned out to be plagiarised (see attachment).
I read the paper thoroughly and found it very confusing: there are many topics included both in the introduction and in the discussion without an apparent logical thread.
The introduction mentions HRV, the importance of diet, inflammation and symbiotic use. Several topics are also mentioned in the discussion but it is never explained how and why the use of a symbiotic affects HRV or body composition.
The methodological approach is also poor. The groups are small in number and the comparison between a sedentary person and an athlete is made in a completely arbitrary manner. The dietary part was not managed, which is clearly a fundamental aspect. A table with baseline data and statistics would be very useful to assess any initial differences between the groups. The only table presented is difficult to read.
https://www.mdpi.com/2072-6643/13/4/1321/htm

Author Response
Dear reviewer, thank you very much for your time in reviewing our manuscript, we are sure that your criticism will improve the paper, and we would like to respond to all your comments.
Aim of this RCT is to identify the effect of a synbiotic in athletes and sedentary people, and their potential varying responses regarding the immune system, autonomic regulation and body composition. Many studies have been published on the subject with poor results. The same authors have also recently published a paper very similar to this one. Many parts turned out to be plagiarised (see attachment).
First of all, we would like to clarify that we do not consider plagiarism because most of the coincidences found belong to part of this research, that is, they use a common methodology, as well as in the description of the figures, because they are different variables and coincide, as stated in your attached document, with an article of our authorship, so we do not consider that it can be considered plagiarism in itself.
I read the paper thoroughly and found it very confusing: there are many topics included both in the introduction and in the discussion without an apparent logical thread. The introduction mentions HRV, the importance of diet, inflammation and symbiotic use. Several topics are also mentioned in the discussion but it is never explained how and why the use of a symbiotic affects HRV or body composition.
In our opinion, the main objective of our study was to test whether the use of a synbiotic could really affect heart rate variability. The relationship between the autonomic nervous system and the immune system is decisive in our study, since many studies have shown that the use of probiotics and synbiotics affect the modulation of the intestinal microbiota through the increase of beneficial bacteria, and thus, an improvement of the immune system. Other diseases, such as obesity, seem to be related to the pathogens of the intestinal microbiota, being synbiotics a possible nutritional tool in their prevention.
We have completed the next paragraph in the introduction of the manuscript to clarify how the use of a synbiotic might affect heart rate variability and body composition:
“Moreover, some reviews argue that the consumption of probiotics, prebiotics and syn-biotics could be effective in improving the performance of athletes by maintaining gastrointestinal and immune function, thus reducing the susceptibility to illness [15–17], as well as improving some metabolic parameters due to changes in the microbiota composition and consequently in body composition [18]”.
“Inflammation is part of the immune response; and inflammatory responses could trigger a wide variety of diseases. Because of the inflammatory response is related to the regulation of the autonomic nervous system (ANS), the biomarker HRV is given special attention in this study, as well as the importance of cytokines activation to prevent the release of inflammatory products into the blood stream [19]. Therefore, synbiotics are proposed as a nutritional strategy for the improvement of the immune system, and as a consequence, possible improvements in the cardiovascular system”.
The methodological approach is also poor. The groups are small in number and the comparison between a sedentary person and an athlete is made in a completely arbitrary manner. The dietary part was not managed, which is clearly a fundamental aspect.
In response to your comment, we would like to clarify that the groups were made according to basic inclusion and exclusion criteria, which were not arbitrary. It is true that the sample is small when dividing the groups, which is why we have included the following sentence in the limitations of the study at the end of the manuscript:
“One of the most important limitations of the study is the low sample size in each group. Since convenience sampling was used in order to use a sample of high-level athletes, the sample of sedentary subjects was obtained from the sample of athletes in order to have the same number of participants. Therefore, an important limitation of this study was not to use a calculation to obtain a representative sample based on the total population, which would have resulted in a higher statistical power.”
In our study, the subjects had to respond, two weeks before the investigation that they were not consuming any type of supplement or medication that could interfere with the consumption of the synbiotic, as well as during the research period. Two subjects were excluded because they were taking antibiotics at the time of the study. In addition, the players had a nutritionist who ensured that their diet was controlled. That is why, we have clarified the management of the diet followed by the participants during the study protocol:
“Participants were asked to follow their regular diet two weeks prior to the investigation and during the protocol. The soccer players followed the diet prepared by their nutritionist”.
A table with baseline data and statistics would be very useful to assess any initial differences between the groups. The only table presented is difficult to read
This information is already shown in figures and tables. The time point PRE refers to baseline values and this is shown for both athletes and sedentary samples in the Placebo and experimental group.
Reviewer 3 Report
The authors tested the effect of a commercial synbiotic preparation on multiple outcomes in athletes and sedentary people. The rationale for the study is not adequately described.
English errors are present throughout the manuscript and extensive editing is required.
Introduction: A stronger explanation is needed as to why a synbiotic would affect HRV. The reason for testing for an effect is lacking.
Methods:
Participant recruitment methods need to be described.
The randomization process needs explanation.
Was a power analysis conducted to determine sample size and the ability to detect main and interaction effects?
Results (mean age, body mass...) belong in the results section within a table.
Synbiotic: ≥ 1x109 (express this as an exponent)
"Among the probiotic strains" - are strains other than those listed in the treatment? What is the proportion of each in the treatment? How was it stored - what temperature? Was viability tested/confirmed?
0.75μg of vitamin (vitamin what?)
Experimental design: Explain the three parties who were blinded to what experimental variables.
Blood samples: Were the participants fasted? How long?
Results: The tables should appear before the figures, in order of presentation of the results in the text.
Æ— p<0.05 - suggest using a different, more easily identifiable, symbol such as *.
Figures: show symbols where significant differences are present.
Discussion:
Line 276: Is BMI reported in table 2?
Line 318: Explain your thinking. Why would the protocol promote weight gain in athletes?
Again, the rationale for testing the effects of the treatment on multiple outcomes is not clear. The potential to find an effect across many outcomes is likely, simply due to chance.
Author Response
First of all, we would like to thank you for your effort and dedication in the revision of our manuscript. We are convinced that with your suggestions it will be substantially improved.
The authors tested the effect of a commercial synbiotic preparation on multiple outcomes in athletes and sedentary people. The rationale for the study is not adequately described. Introduction: A stronger explanation is needed as to why a synbiotic would affect HRV. The reason for testing for an effect is lacking
In our opinion, the main objective of our study was to test whether the use of a synbiotic could really affect heart rate variability. The relationship between the autonomic nervous system and the immune system is decisive in our study, since many studies have shown that the use of probiotics and synbiotics affect the modulation of the intestinal microbiota through the increase of beneficial bacteria, and thus, an improvement of the immune system. Other diseases, such as obesity, seem to be related to the pathogens of the intestinal microbiota, being synbiotics a possible nutritional tool in their prevention.
We have completed the next paragraph in the introduction of the manuscript to clarify how the use of a synbiotic might affect heart rate variability and body composition:
“Moreover, some reviews argue that the consumption of probiotics, prebiotics and syn-biotics could be effective in improving the performance of athletes by maintaining gastrointestinal and immune function, thus reducing the susceptibility to illness [15–17], as well as improving some metabolic parameters due to changes in the microbiota composition and consequently in body composition [18]”.
“Inflammation is part of the immune response; and inflammatory responses could trigger a wide variety of diseases. Because of the inflammatory response is related to the regulation of the autonomic nervous system (ANS), the biomarker HRV is given special attention in this study, as well as the importance of cytokines activation to prevent the release of inflammatory products into the blood stream [19]. Therefore, synbiotics are proposed as a nutritional strategy for the improvement of the immune system, and as a consequence, possible improvements in the cardiovascular system”.
English errors are present throughout the manuscript and extensive editing is required.
The article has been reviewed by a native English-speaker colleague.
Methods:
Participant recruitment methods need to be described and the randomization process needs explanation
All the athletic subjects belonged to the same football team in the third division of the Spanish football league, while the sedentary subjects were sedentary university students (less than 150 minutes per week of physical activity).
Each subject was assigned a code (sedentary or athlete), as well as each of the tests (HR variability, anthropometry and blood samples) according to when they were performed (baseline or post) and according to the treatment with which each subject was administered (placebo or synbiotic). These codes, and therefore the samples, did not allow the investigator to know the group to which they belonged during the determination of the experimental variables and their processing, thus completing the triple-blind study.
The following paragraph has been included in materials and methods to clarify the randomization process:
“Random assignment through the use of coding provided by the laboratory allowed researchers and participants to be blinded to the treatment provided. The evaluation and analysis was also blinded, thus completing the triple-blind study”.
Was a power analysis conducted to determine sample size and the ability to detect main and interaction effects?
We did not calculate de sample size. We are aware of this limitation, therefore we have included a limitations paragraph in the discussion section explaining why we did not calculate the representative sample size since we used a convenience sampling. The information described is:
“Since convenience sampling was used in order to use a sample of high-level athletes, the sample of sedentary subjects was obtained from the sample of athletes in order to have the same number of participants. Therefore, an important limitation of this study was not to use a calculation to obtain a representative sample based on the total population, which would have resulted in a higher statistical power.”
Results (mean age, body mass...) belong in the results section within a table
The descriptive data of the sample has been moved to the results section as suggested by reviewer #3 as well as introduced within a table instead of in text format.
Synbiotic: ≥ 1x109 (express this as an exponent)
Thank you for your appreciation, we have made the change.
"Among the probiotic strains" - are strains other than those listed in the treatment? What is the proportion of each in the treatment? How was it stored - what temperature? Was viability tested/confirmed?
The synbiotic was provided by the laboratory, each stick being composed of the following amount per recommended daily dose:
Bifidobacterium lactis, Lactobacillus rhamnosus and Bifidobacterium longum ES1( ≥ 1x109).
Fructooligosaccharides (200mg)
Zinc Sulfate (4,12mg equivalent to 1,5mg of Zinc)
Cholecalciferol (0,75µg of Vitamin D)
Sodium Selenite (20µg equivalent to 8,25µg of Selenium)
The product should be kept in a cool and dry place. The product is stable at room temperature. Moreover, the viability of the product was previously tested by Heel España S.A.U. laboratories.
We have included all the information suggested in the materials and methods section.
0.75μg of vitamin (vitamin what?)
We have corrected the error as we had forgotten to include the specific type of vitamin (D), we have included this in the manuscript.
Experimental design: Explain the three parties who were blinded to what experimental variables.
The three blinded parties in this research were: the participants, who were unaware of the treatment they were receiving, the researchers, as well as the analysis and evaluation of the groups which was done without knowing the identity of the participants. This was done for all variables in the study.
Blood samples: Were the participants fasted? How long?
Thank you for the commentary. We have included the next sentence in the experimental design section to clarify the fasting that the participants had to perform prior to sample collection.
The procedures and materials of the tests were performed in the same way for the “baseline-tests” and “final-tests” to alter each of the measurements as little as possible and participants had to come fasting for at least 12 hours prior to sampling.
Results: The tables should appear before the figures, in order of presentation of the results in the text.
We have changed the order. Tables appear now first and figures at the end of the results section.
Æ— p<0.05 - suggest using a different, more easily identifiable, symbol such as *. Figures: show symbols where significant differences are present.
Thank you for the advice. We have now used *.
Discussion:
Line 276: Is BMI reported in table 2?
We didn’t report changes in BMI since the height of the subjects does not change. Thus, the use of BMI would be exactly the same value and same change as body mass.
Line 318: Explain your thinking. Why would the protocol promote weight gain in athletes?
In our opinion, the possible increase in body weight, especially in athletic subjects, is due to the increase in muscle mass. This could be due to the treatment or to the performance of the sport itself, as stated in the discussion of this manuscript.
Round 2
Reviewer 1 Report
The authors have addressed the issues raised by me
Author Response
The authors have addressed the issues raised by me.
Thank you for your efforts in reviewing our manuscript. We are sure that your suggestions in the last revision have improved the quality of our work.
Reviewer 2 Report
The authors have made some minor changes to the paper.
Unfortunately, the basic shortcomings remain. There are still many plagiarised paragraphs. This is an indisputable fact.
This is a small sample study showing that a synbiotic has no effect on the immune system, autonomic regulation and body composition in subjects whose diets were not monitored.
The authors should simplify the work and above all specify in both the title and the conclusions that a synbiotic has no positive effect on the outcomes considered.

Author Response
The authors have made some minor changes to the paper.
Unfortunately, the basic shortcomings remain. There are still many plagiarised paragraphs. This is an indisputable fact.
We have included the next sentence in the experimental design section to clarify that this study is part of a larger study with the same sample.
“This research is part of a larger study and there is a previously published article with the same sample [20]”.
Therefore, we would like to point out that most of the coincidences are anecdotal because groups of words are repeated, however we have modified some of the parts with more coincidences with our previous publication, which is why we did not consider plagiarism.
Abstract: “A synbiotic (Gasteel Plus®, Heel España S.A.U.) comprising a blend of probiotic strains including Bifidobacterium lactis CBP-001010, Lactobacillus rhamnosus CNCM I-4036, and Bifidobacterium longum ES1, was administered to the experimental group and a placebo was given to the control group for 30 days”.
Introduction:
“Furthermore, some studies suggest that taking probiotics, prebiotics, and synbiotics can help athletes perform better by maintaining gastrointestinal and immunological function, lowering their susceptibility to disease [15–17]”.
Experimental design:
“The main goal of this one-month, triple-blind, randomized, placebo-controlled pilot trial was to determine whether there were any differences in the effects of the synbiotic Gasteel Plus® supplementation between sedentary people and athletes”.
“All participants were informed about the study two weeks before the intervention and asked to provide written informed consent before participating in the study, which had previously been approved by the ethics committee of the Catholic University of Murcia (Spain) in accordance with current legislation (CE031810). ClinicalTri-als.gov was used to register this trial (identifier: NCT04776772: available from web-site)”.
Blood samples:
“Blood samples were taken from the individuals at 8 a.m. and deposited into col-lection tubes containing the anticoagulant EDTA and coagulating agents, respectively, to isolate plasma and serum. The plasma and serum were centrifuged for 10 minutes at 1600 and 1800 x g, respectively. As serum and plasma samples were collected, they were tagged and gradually refrigerated at -20°C. Finally, samples were kept at -80°C until they could be analyzed”.
“The LuminexTM 200 System instrument (Luminex Corporation, Austin, Texas, USA) was utilized to determine the examined cytokines, Interleukin 6 (IL-6), Interleukin 8 (IL-8) and Tumor necrosis factor alpha TNF-α), using the ProcartaPlex TM Multiplex Immunoassay. The procedures were carried out according to the manufacturers' instructions, and the results were quantified using an ELISA auto analyzer (Sunrise, Tecan, Männendorf, Switzerland)”.
Statistical Analysis:
“The SPSS for Windows statistical tool was used for data collection, treatment, and analysis (version 20.0; SPSS, Inc., Chicago, IL, USA). The mean and SD of descriptive statistics were calculated. The Shapiro-Wilks test was used to check the assumption of normality before performing parametric testing”.
Figure legend:
Figure 1. The effect of training and a synbiotic on the IL-6 cytokine. A) Baseline serum IL-6 con-centrations in sedentary men (n=14) and athletes (n=13); B) Training effects on synbiotic effects on serum IL-6 concentration (n=7 and n=6 in sedentary and athlete groups, respectively); C) Effect of synbiotic consumption on IL-6 in sedentary individuals with placebo (n=7) or synbiotic (n=7); D) Effect of synbiotic consumption on IL-6 in athlete individuals with placebo (n=6). The determina-tions are expressed by the mean ± SD of each of the samples. No significant differences were seen between groups (p>0.05).
Figure 2. The effect of training and a synbiotic on the IL-8 cytokine. A) Baseline serum IL-8 con-centrations in sedentary men (n=14) and athletes (n=13); B) Training effects on synbiotic effects on serum IL-8 concentration (n=7 and n=6 in sedentary and athlete groups, respectively); C) Effect of synbiotic consumption on IL-8 in sedentary individuals with placebo (n=7) or synbiotic (n=7); D) Effect of synbiotic consumption on IL-8 in athlete individuals with placebo (n=6). The determina-tions are expressed by the mean ± SD of each of the samples. * p <0.05 compared to basal.
Figure 3. The effect of training and a synbiotic on the TNF-α cytokine. A) Baseline serum TNF-α concentrations in sedentary men (n=14) and athletes (n=13); B) Training effects on synbiotic effects on serum TNF-α concentration (n=7 and n=6 in sedentary and athlete groups, respectively); C) Effect of synbiotic consumption on TNF-α in sedentary individuals with placebo (n=7) or synbiotic (n=7); D) Effect of synbiotic consumption on TNF-α in athlete individuals with placebo (n=6). The determinations are expressed by the mean ± SD of each of the samples. No significant differences were seen between groups (p>0.05).
This is a small sample study showing that a synbiotic has no effect on the immune system, autonomic regulation and body composition in subjects whose diets were not monitored. The authors should simplify the work and above all specify in both the title and the conclusions that a synbiotic has no positive effect on the outcomes considered.
We have changed the title of the manuscript to indicate that there is no effect after the consumption of our synbiotic and this fact is already stated in the conclusion section.
Reviewer 3 Report
Are the subjects in this paper the same as those in:
Quero, C.D.; Manonelles, P.; Fernández, M.; Abellán-Aynés, O.; López-Plaza, D.; Andreu-Caravaca, L.; Hinchado, M.D.; Gálvez, I.; Ortega, E. Differential Health Effects on Inflammatory, Immunological and Stress Parameters in Professional Soccer Players and Sedentary Individuals after Consuming a Synbiotic. A Triple-Blinded, Randomized, Placebo-Controlled Pilot Study. Nutrients 2021, 13, 1321. https://doi.org/10.3390/nu13041321
If so, these results should have been included in that paper. Also, if so, use the same descriptor - soccer or football players.
Why are results not posted on the Clinical Trials site?
At least the prior experiment should be discussed and shown to be substantially unique from this one.
The link between HRV and synbiotic impact remains poorly explained. One 2002 paper was added as a reference.
Line 110: "The soccer players followed the diet prepared by their nutritionist." If the sedentary group did not, this is an important confounding variable in the experiment. Diet needs to be controlled for both, not just one, group.
Table 1. State the number of subjects for each group.
Figure 1 is missing results of significance tests.
Legends for Figures 2 & 3 are not located properly (lines 232-245). The significance symbol in Figure 2 does not match that in the legend.
Author Response
Thanks for the comments and suggestions to improve the quality of out work
Are the subjects in this paper the same as those in:
Quero, C.D.; Manonelles, P.; Fernández, M.; Abellán-Aynés, O.; López-Plaza, D.; Andreu-Caravaca, L.; Hinchado, M.D.; Gálvez, I.; Ortega, E. Differential Health Effects on Inflammatory, Immunological and Stress Parameters in Professional Soccer Players and Sedentary Individuals after Consuming a Synbiotic. A Triple-Blinded, Randomized, Placebo-Controlled Pilot Study. Nutrients 2021, 13, 1321. https://doi.org/10.3390/nu13041321
If so, these results should have been included in that paper. Also, if so, use the same descriptor - soccer or football players.
We were not able to introduce all the variables in our previous paper as it would be too long and not suitable for publication because of its length.
Due to participants are the same as in the previous research, we have changed the term football players to soccer players.
Why are results not posted on the Clinical Trials site?
Results are not posted on the clinical Trial site because the registration was done after the measurements were completed but the final results were not available yet. Moreover, it was not mandatory, so we did not consider it a significant fact.
At least the prior experiment should be discussed and shown to be substantially unique from this one.
We have included the next sentence in the experimental design section to clarify that this study is part of a larger study with the same sample.
“This research is part of a larger study and there is a previously published article with the same sample [20]”.
The link between HRV and synbiotic impact remains poorly explained. One 2002 paper was added as a reference
Unfortunately, there is no previous research analyzing specifically the effect of synbiotic consumption on HRV. That is why we have not been able to state a direct relationship between these two facts. On the other hand, we have tried to explain which can theoretically be the effect of synbiotic consumption on HRV in the introduction section.
Line 110: "The soccer players followed the diet prepared by their nutritionist." If the sedentary group did not, this is an important confounding variable in the experiment. Diet needs to be controlled for both, not just one, group
The sedentary group did not follow a specific diet because the necessary resources were not available. However, as specified in the article, during the protocol, they were not allowed to consume any probiotic, prebiotic or fermented products (yogurt or other foods), as well as not taking any type of medication or supplement that could interfere with the results of the study.
Considering this fact as a possible confounding factor, we have included the following sentence in the limitations of the study:
Another important limitation was that it was not possible to control the diet of the sedentary sample, who followed their normal daily diet.
Table 1. State the number of subjects for each group
Sorry for the inconvenience. Numbers of subjects are now included in table 1.
Figure 1 is missing results of significance tests
Statistics are shown in figures 1, 2 and 3 but significant differences were not seen. This is the reason why symbols are not used in some of the figures. However as you suggested, we have included the following sentence in figures 1 and 3:
‘’No significant differences were seen between groups (p>0.05)’’.
Legends for Figures 2 & 3 are not located properly (lines 232-245). The significance symbol in Figure 2 does not match that in the legend.
Thank you for the appreciation, we have made the correction.